# Effectiveness of digital technology interventions to reduce loneliness in adults: a protocol for a systematic review and meta-analysis

Syed Ghulam Sarwar Shah ,[1,2] David Nogueras,[3] Hugo van Woerden,[4,5] Vasiliki Kiparoglou[1,6]

¹NIHR Oxford Biomedical Research Centre, Oxford University Hospitals NHS Foundation Trust, Oxford, UK
²Radcliffe Department of Medicine, University of Oxford, Oxford, UK
³DNM Consulting, London, UK
⁴NHS Highland, Assynt House, Inverness, UK
⁵Centre for Health and Science, University of the Highlands and Islands, Inverness, UK
⁶Nuffield Department of Primary Care Health Sciences, University of Oxford, Oxford, UK

**Correspondence to**
Dr Syed Ghulam Sarwar Shah; Sarwar.Shah@ouh.nhs.uk

## ABSTRACT

**Introduction** Loneliness is an emerging public health problem that is associated with social, emotional, mental and physical health issues. The application of digital technology (DT) interventions to reduce loneliness has significantly increased in the recent years. The effectiveness of DT interventions needs to be assessed systematically.

**Methods and analysis** Aim: To undertake a systematic review and meta-analysis on the effectiveness of DT interventions to reduce loneliness among adults.
Design: Systematic review and meta-analysis.
Data sources: PubMed, Medline, CINAHL, EMBASE and Web of Science.
Publication period: 1 January 2010 to 31 July 2019.
Inclusion criteria: Primary studies involving the application of DT interventions to reduce loneliness, involving adult participants (aged ≥18 years), follow-up period ≥3 months and published in the English language.
Synthesis and meta-analysis: A narrative summary of the characteristics of included studies, findings by the type of DT intervention, and the age, gender and ethnicity of participants. A meta-analysis by the study design and duration of follow-up and determination of random effects size using the RevMan V.5 software.
Quality of evidence and bias: Quality of evidence assessed the RoB V.2.0 (revised tool for Risk of Bias in randomized trials) and ROBINS-I (Risk Of Bias in Non-randomized Studies—of Interventions) tools for randomised control trials and non-randomised studies, respectively. Heterogeneity between studies will be determined by the $I^2$ and Cochran's Q statistics and publication bias checked with funnel plots and the Egger's test.

**Ethics and dissemination** Ethics approval was not required for this protocol. The findings will be disseminated through journal articles and conference presentations.

**PROSPERO registration number** CRD42019131524

## INTRODUCTION
### Definitions
In the literature, loneliness and social isolation are often reported together but they differ from each other.[1] Loneliness is defined as a subjective feeling that arises when an individual perceives a descrepancy between the actual and the desired social relationships.[2 3] While, social isolation is an objective feeling,[3] which develops due to the absence of social contact with the family, friends,[1] individuals and society.[3] In addition, loneliness and social isolation have distinct pathways to adverse health effects.[4 5] Loneliness creates pain and distress[3] and it is associated with adverse health effects.[1]

### Burden of loneliness
Loneliness is increasing, especially in developed countries[6] such as Australia,[7] Japan,[8] the UK[9] and the USA.[10] Loneliness is seen as an epidemic[5] and a rising public health problem[11] leading to social, mental and physical health problems.[12–14]

### Loneliness and demographic factors
Loneliness can affect individuals of any age and members of any community.[3] It is prevalent in young children and adolescents[15] who are more susceptible to loneliness as they go through social and personal transformations.[16] In children and adolescents,

loneliness is associated with chronic physical conditions.[17] In adults, loneliness is associated with female gender, older age, inadequate income, lower educational level, living alone, low quality of social relationships, poor self-reported health, poor functional status, marital status (unmarried)[18] including people who have never been married, are widowed or divorced.[19] In older people, loneliness is common because they are more vulnerable because of age-related changes and losses.[20]

**Strengths and limitations of this study**

► The main strength of this study includes a systematic assessment of evidence on the effectiveness of digital technology interventions to reduce loneliness, which is imperative from the health and social care and public health perspectives.
► Another strength of the study is the involvement of two independent researchers (and a third researcher as an arbitrator) involved in the identification, screening and inclusion of studies and data extraction on a predefined template using the population, intervention(s), comparator(s) and outcome(s) framework.
► Limitations may include missing identification of additional relevant studies because of search filters such as the English language.

### Loneliness and physical and mental health

Loneliness is associated with physical health and mental health risks.[20] Loneliness is also positively associated with chronic conditions such as cancer[19] and cardiovascular disease[21] and has a negative influence on the quality of life, health and survival.[22] In old age, loneliness is associated with reduced quality of life, poor health, maladaptive behaviour and depressed mood[23] as well as increased risk of developing Alzheimer's disease.[24]

Loneliness is associated with mental health conditions such as depression, low self-esteem, anxiety, perceived stress,[16 25] psychosis,[26] shame and fear,[18] incident dementia,[27] sleep disorders[28] and suicidal thoughts in older adults.[29]

Thus, loneliness increases risks not only to physical health[5 30] and mental health[12–14] but also increases risk of premature mortality and all-cause mortality[5] especially in older adults.[31]

### Loneliness and social and physical environment

Loneliness is associated with social and physical environment such as boredom and inactivity, recent losses of family and friends, inaccessible housing, inadequate resources for socialising, unsafe neighbourhoods, migration patterns,[18] with lack of psychological or social support,[19] low economic level and living arrangements such as living alone.[32]

More importantly, not only are the numbers of adults at the risk of loneliness rising but also the costs associated with loneliness are also increasing.[33] To address this double-edged sword, tackling loneliness is important[1] through effective interventions and strategies.

### Interventions to tackle loneliness

Loneliness could be tackled with various interventions,[33] broadly divided into two categories, that is, social interventions and technological interventions.

Social interventions applied to reduce loneliness include befriending, residential and school-based camps, reminiscence therapy, animal interventions, gardening, physical activity and technology.[34] Interventions focusing on social network maintenance and enhancement have also been applied and found to be useful to combat loneliness.[20] Social interventions could be combined with the application of technology such as online peer-to-peer interactions and support groups through social media platforms to alleviate loneliness especially in persons with psychotic disorders.[35] However, loneliness in older people

can create serious problems that could not be alleviated with the social support only[32]; other types of interventions are required such as technological interventions (eg, digital applications (apps), online social networks and social robots) to enhance emotional support and social interaction.[36]

### Digital technology interventions

The term 'digital technology' (DT) refers to the technology, equipment and applications that process information in the form of numeric codes, usually a binary code, which is processed by many devices such as computers, smartphones and robots.[37] Research and development in DT has become essential alongside social change and ubiquity of computer technologies, which are an integral part of the daily life of many people.[38] Many kinds of technological interventions could be applied to reduce loneliness.[34]

We therefore focus on DT interventions to reduce loneliness. We will assess any intervention that involves the application of DT to reduce and alleviate loneliness in the adult population.

### Previous systematic reviews on technological interventions for loneliness

There are some systematic reviews on technological interventions for tackling loneliness. For example, Pearce *et al* undertook a systematic review on the availability and use of robotics by older people and reported that robotic technologies could support older people and those with disabilities in independent living; however, they suggested that there was a need for further research to achieve the full potential of robotic technologies on social connectedness.[39] In addition, a meta-analysis on the effectiveness of computer and internet training interventions intended to reduce loneliness and depression in older adults by Choi *et al* reported that computer and internet programme were effective in managing loneliness among older adults.[40]

Morris *et al* conducted a systematic review of literature on the effectiveness of smart-home technologies, that is, passive sensors, monitoring devices, robotics and environmental control systems for promoting independence, health, well-being and quality of life, in older adults.[41] In another systematic review Morris *et al* evaluated the effectiveness of tailored internet programme using computers and the internet, which they called as smart technologies and found that digital /smart technological interventions have positive outcomes in enhancing social connectedness in older people compared with traditional social care interventions.[42] A literature review by Hagan *et al* investigated the effectiveness of social therapeutic interventions to reduce loneliness in older people and found a significant reduction in loneliness.[43] A systematic review on the effectiveness of virtual reality and online games on enabling physical activity in older peoples living at home by Miller *et al* reported a high risk of bias and weakness in the studies included in their review.[44]

## Rational

Technology can provide opportunities for social connectedness and thus help in reducing loneliness in older adults; however, studies involving technological interventions to alleviate loneliness in frail and institutionalised older adults are limited.[45]

For example, a systematic review by Pearce *et al* focused only on robotic technologies and their effectiveness in independent living by older people and suggested that the effect of robotics on older peoples' safety and social connectedness needs further research.[39] These findings suggest that assessment of the effect of robotic technologies on loneliness needs to be undertaken.

A systematic review by Morris *et al* involved assessment of the effectiveness of smart-home technologies: passive sensors, monitoring devices, robotics and environmental control systems for promoting independence, health, well-being and quality of life in older adults; however, this review was limited in finding only one study on the effectiveness of smart-home technologies and the focus of their review was not on loneliness but on older people's independent living in homes.[41]

Another systematic review by Morris *et al* on smart technologies for improving or maintaining social connectedness also has limitations such as a narrow definition of smart technologies, which they searched as 'computers' and the 'internet' and combined with assistive technologies.[42] Other limitations in the later systematic review by Morris *et al*[42] include the focus on social connectedness and older people living in homes, and inclusion of studies published in only 3 years period from January 2010 to January 2013. Because of these limitations, they called for further research regarding smart technologies for reducing loneliness in older adults.[42]

We anticipate the publication of new research studies from 2013 to the present and a need to assess the latest research involving digital technologies to reduce loneliness in older people. This is important because the technology is evolving very rapidly and with new technologies (eg, smart speakers, new sensors, new robot versions and so on) being developed and becoming available and adopted very rapidly. Therefore, we need to assess their potential impact on adults with loneliness.

The evidence for technology-assisted interventions to reduce loneliness and the effects on the social health and well-being of older people is limited.[23] Moreover, the latest research interventions involving newer digital technologies need to be assessed.[43] It is also necessary to identify technological interventions that are effective in reducing loneliness[23] and identify how technological innovations can be promoted, marketed and implemented to benefit older people with loneliness.[42]

## SYSTEMATIC REVIEW REGISTRATION

This protocol was registered with the PROSPERO database (www.crd.york.ac.uk/prospero/), which is an International prospective register of systematic reviews[46] on 10 June 2019 with a registration ID of PROSPERO 2019[47] and it can be accessed online at http://www.crd.york.ac.uk/PROSPERO/display_record.php?ID=CRD42019131524.

## METHODS AND ANALYSIS

For the development and writing of this protocol, we will follow the Joanna Briggs Institute's guidelines and template for writing a protocol for a systematic review of effectiveness evidence and meta-analysis[48] and the Preferred Reporting Items for Systematic Review and Meta-Analysis Protocols 2015 checklist,[49] which is given as appendix I.

### Aims and objectives

The aim and objectives of this study are as follows.

#### Aim

1. To undertake a systematic review and meta-analysis on the effectiveness of DT interventions to reduce loneliness among adults.

#### Objectives

1. To identify DT interventions used to reduce loneliness in adults.
2. To assess the effectiveness of DT interventions to reduce loneliness in adults.

### Review question(s)

Primary review question: How effective are DT interventions in reducing loneliness in adults?

Secondary review question: What DT interventions are used for tackling loneliness in adults?

### Main outcome measure

Loneliness will be the main outcome measure in our study.

#### Timing and effect measures

We will include preintervention and postintervention measurements of loneliness using any of the following loneliness measures: UCLA Loneliness Scale, De Jong Gierveld 6-Item Loneliness Scale, Campaign to End Loneliness Measurement Tool and any other loneliness scale (eg, single-item questions, also known as self-rating measures of loneliness).[50] We will include studies that will have a follow-up period of at least 3 months or more to measure the outcome(s). We will extract information on the measurement of loneliness at the baseline (before the intervention) and every follow-up measurement for the intervention group and control group, if any, depending on the design of the studies included in our systematic review. In the case of more than one follow-up measurement, we will run a series of meta-analyses as explained in the Meta-analysis section.

#### Inclusion criteria

We will include studies that meet a predefined set of inclusion criteria (table 1).

**Table 1** Inclusion criteria

| Parameter | Inclusion criteria |
|---|---|
| Condition | Loneliness |
| Publication dates | From 1 January 2010 to 31 July 2019 |
| Publication types | Primary research published as journal articles |
| Study types/designs | *Primary research:* RCTs/clinical trials, observational studies (cohort (before and after) and case–control studies) |
| Study subjects/participants | Humans—adults |
| Age | 18 years and above |
| Gender | Male and female |
| Interventions | Digital technology interventions, including application of sensors, (social) robots, Internet, social media, (smart) phones, online tools, iPads, computers and tablets, world wide web, videos and online chats, groups, meetings, conferences and messages |
| Outcome(s) | Loneliness |
| Follow-up time | Three months or more |
| Language | English |
| Research disciplines | Public health and social care |
| Geographic location/country of study | All countries |
| Settings/context | Residential dwellings including private residences and care homes/nursing homes |

## Data sources

We will systematically search for articles published in five large and widely used online bibliographic databases: PubMed, CINAHL, Web of Science, Medline and EMBASE (Excerpta Medica dataBASE), which cover literature in the fields of health sciences, medicine, nursing, allied health, biomedicine, health technology and healthcare. Literature searches through these five databases will cover publication period from 1 January 2010 to 31 July 2019.

We will also review the references of shortlisted articles for identifying any relevant studies. We will write to the authors for full copies of any articles that could not be accessed or retrieved full via the Bodleian Health Care Libraries, University of Oxford.

## Search strategy and parameters
### Preparation of a list of keywords
*Preliminary literature searches*
Initially, a preliminary literature search of the PubMed database was carried out using a set of keywords (Box 1).

As a result of our preliminary search query, we captured 100 articles, which were reviewed independently by two reviewers (SGSS and DN), who screened titles of all these articles and recommended relevant articles to be included in the second stage of screening, that is, reading the abstracts.

*List of keywords*
We noted subject terms reported in all articles that the two reviewers recommended for the second stage. We reviewed these keywords and prepared a refined list of keywords. We noted that there were very few keywords on

the various types of digital tools and technologies that we thought could be relevant for identifying digital interventions for addressing the issue of loneliness. Therefore, we divided keywords into two categories, that is, conditions/issues and intervention/technology (table 2). A full list of keywords is shown in table 2 and these will be used for our full literature searches and the identification of relevant articles. We will search for these keywords in the titles, abstracts and author keywords fields in the selected databases.

---

**Box 1    Preliminary search user query and filters applied in PubMed**

**User Query**

Loneliness[MeSHMajor Topic] AND needs AND intervention* AND ((systematic[sb] OR Review[ptyp] OR Randomized Controlled Trial[ptyp] OR ObservationalStudy[ptyp] OR Meta-Analysis[ptyp] OR Journal Article[ptyp] OR Evaluation Studies[ptyp] OR Clinical Trial[ptyp]) AND ("2010/01/01"[PDat]: "2018/11/12"[PDat]) AND Humans[Mesh] AND English[lang] AND (Female[MeSH Terms] ORMale[MeSH Terms]) AND (jsubsetn[text] OR medline[sb]) AND (aged[MeSH] OR middle age[MeSH] OR adult[MeSH:noexp] OR adult[MeSH] OR adolescent[MeSH] OR child[MeSH:noexp]))

**Filters activated**

Systematic Reviews, Review, Randomized Controlled Trial, Observational Study, Meta-Analysis, Journal Article, Evaluation Studies, Clinical Trial, Publication date from 2010/01/01 to 2018/11/12, Humans, English, Female, Male, Nursing journals, MEDLINE, Aged: 65+ years, Middle Aged: 45-64 years, Adult: 19-44 years, Adult: 19+ years, Adolescent: 13-18 years, Child: 6-12 years, Field: Title/Abstract

---

**Table 2** List of keywords

| Condition / issue | Intervention / technology |
|---|---|
| Loneliness | Digital |
| Lonely | Technology |
| Isolation | Sensor* |
| Aloneness | Robot* |
| Disconnect* | Internet |
| Solitude | Social media |
| Singleness* | *Phone* |
| Lonesomeness | Online |
| Solitariness | IPad |
| Remoteness | Tablet |
| | Computer* |
| | Electronic |
| | Web |
| | Video |
| | Video conference |

**Box 2    Data extraction form**

1. Authors (and year of publication).
2. Country of study.
3. Aim/objectives of the study.
4. Research design.
5. Settings.
6. Characteristics of participants (age, gender and ethnicity).
7. Health/medical condition.
8. Sampling method.
9. Sample size.
10. Participant attrition (numbers / %).
11. Research method(s) / data collection tool(s).
12. Intervention(s), for example, type of digital technology.
13. Comparator(s), for example, alternative intervention or placebo/care as usual.
14. Total duration of the intervention (weeks / months).
15. Measurement stages, for example, baseline, follow-up 1, 2, 3 (weeks/months after the baseline).
16. Outcome/results/findings (including the statistics, eg, mean values, SD and CIs).
17. Authors' conclusion(s).
18. Quality of study (reviewers' evaluation/remarks about the study).

## Running of literature searches

Systematic literature searches will be undertaken in the selected five online bibliographic databases using the preidentified list of keywords (table 2).

The literature will be searched using the keywords that will be searched in only the 'title' and 'abstract' search fields in the selected bibliographic databases. The keywords will be used first in the 'subject headings' such as Medical Subject Headings major terms in the PubMed or equivalent in other databases.

The searches will be filtered by applying the inclusion criteria (table 1). We will identify literature using the keywords and applying the Boolean operators, that is, 'OR', 'AND' and 'NOT' while searching for literature in different selected electronic/online bibliographic databases, as reported in the Management of study records/references section.

In addition to searching through selected online bibliographic databases, we will search for relevant articles through searching references' lists of all selected articles.

We will seek support from the Bodleian Health Care libraries staff for running literature searches.

## Management of study records/references

We will keep a record of amendments in the protocol, if any, using an Excel spreadsheet. All records found in searches through the selected databases will be directly downloaded and exported to the RefWorks software, which is a web-based bibliography and database manager.[51]

We will have a quick look at all articles to check whether any information is missing ensuring completeness of bibliographic details. Also, we will note the total number of articles downloaded. Thereafter, we will identify and remove duplicate articles using the RefWorks software as well as manually. We will note the total number of duplicate articles and all duplicate articles will be deleted and the total number of articles remaining will be noted again.

Subsequently, we will create a list of all unique articles in an Excel spreadsheet to facilitate articles screening and shortlisting independently by two researchers (SGSS and DN). In addition, we will use an Excel spreadsheet for extracting data from shortlisted articles on a predefined template (Box 2). For citing articles in our papers and publications, we will use the Zotero software, which is an open-source and free research tool for references management and citations.[52]

We will use the Preferred Reporting Items for Systematic Review and Meta-Analysis flow diagram (figure 1) for the identification, screening, eligibility and inclusion of relevant studies and data extraction,[53] as explained in the following sections.

## Selection of articles
### Screening by title

After deleting duplicate articles, we will prepare a list of titles of all articles that will be independently screened by a team of two researchers (SGSS and DN), who will independently determine the suitability of articles for the second stage of screening by the abstract. At this stage, all article titles marked as 'to be excluded' by both researchers will be removed and all articles marked as 'to be included' will be saved to a different file for further screening of the abstract. For articles where the recommendations of both researchers involved in title screening differed from each other, the third reviewer (HvW) will review the articles and have the final say in either including or excluding an article. Thereafter, a list of all included articles will be prepared for the second screening by reading the abstracts.

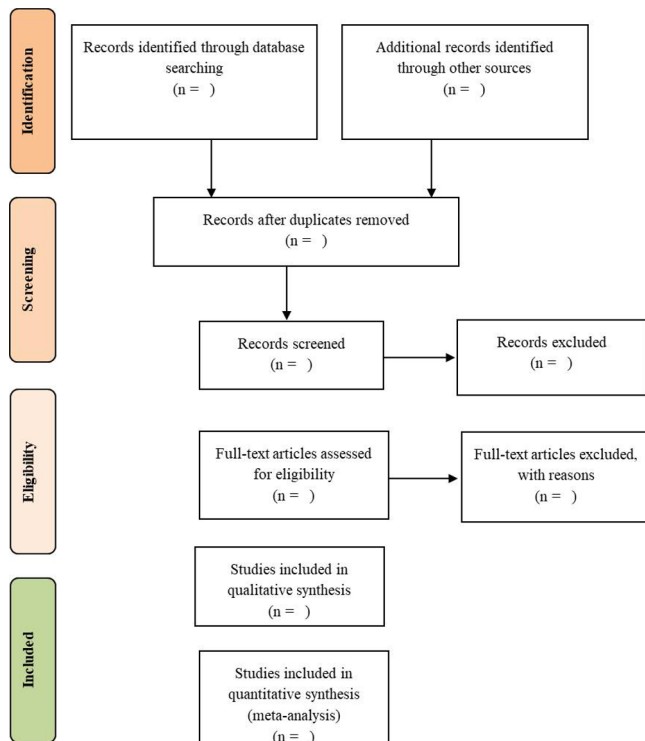

**Figure 1** PRISMA flow diagram. PRISMA, Preferred Reporting Items for Systematic Review and Meta-Analysis.

### Screening by abstract

We will read abstracts of all articles retained after the initial article screening by title. These will be screened further for relevance with respect to the aim and objectives of the study. The screening of abstracts will be undertaken by the same two researchers (SGSS and DN) involved in the title screening. The process for short listing of articles through abstract screening will be the same as reported under the article screening by title stage.

### Screening by reading full-text

We will collect the full text of all articles that will be shortlisted at the abstract level screening. A team of two researchers who were involved in the title and abstract screening stages will review the shortlisted articles and will decide whether an article should be included in or excluded from the study. It might be possible that some articles will be found not relevant to the study aim and objectives; hence, such articles will be excluded from the study; while the remaining articles will be included in the last phase of article selection.

### Selection and inclusion of studies

All of those articles that the two researchers (SGSS and DN) will independently identify as relevant to the study will be included in the systematic literature review and meta-analysis. Any differences between the two researchers will be resolved by the third reviewer (HvW). Thereafter, full text copies of all articles included in the systematic review will be read for data extraction as reported below.

### Data extraction process and items

Two independent reviewers (SGSS and DN) will read the full text of shortlisted articles. The two reviewers will independently extract information from the shortlisted articles on a predetermined data extraction template (Box 2). We will extract data reported for all measurements including measurements at the baseline and all subsequent follow-up stages. Avoidance of bias and reduction of errors in the data extractions is imperative in systematic literature reviews and meta-analysis[49]; hence, after the completion of data extraction, the two data extraction forms will be compared and any differences and discrepancies will be reconciled with discussion and in the case of non-agreement between the two reviewers, a third researcher (HvW) will be involved for arbitration and the final decision.

Since our systematic review will assess the effectiveness of DT interventions; we will therefore extract thorough details of interventions, which is important for the reproducibility of effective interventions.[54 55] We will not extract data at the level of individual patients but the cumulative data from the study, which will be used for the synthesis and inclusion in the meta-analysis as explained below. If data reported in selected studies were found difficult to extract or incomplete, we will attempt to contact the original researchers with regard to data reported in their published articles. Identification of multiple reports and publication of the same data could be possible while undertaking a systematic review.[49] To deal with duplicate or multiple reporting of the same data, we will only report the data once and comprehensively reported in a detailed form.

We will extract data on various items as shown in the data extraction template (Box 2), which we have developed a priori in-house using an Excel spreadsheet. With regard to the outcomes and measures, our primary outcome of interest is loneliness as measured by any of the loneliness scales as reported earlier.

## DATA SYNTHESIS AND REPORTING RESULTS

There are two options for reporting data in a systematic review of effectiveness ie, statistical synthesis (meta-analysis) and narrative summary (narrative synthesis).[48] We will extract and analyse data at the study level. We will report findings using a summary of the characteristics of included studies and the findings by the type of DT intervention, the age, gender and ethnicity of participants, if possible. In addition, we will undertake quantitative synthesis and descriptive/narrative synthesis for quantitative and qualitative studies respectively. Moreover, we will exclude studies with missing values from the meta-analysis.

### Meta-analysis

In the meta-analysis, we will use the effect sizes as measured by common quantitative indicators such as the risk ratio (RR), risk difference (RD) and OR for the

dichotomous outcomes and the weighted mean difference (WMD), and standardised mean difference (SMD) for continuous outcomes.[49] For every study included in our systematic review and meta-analysis, we will calculate the effect size as reported by Masi *et al*.[33] In addition, we will report a statistical synthesis of our meta-analysis using a statistical summary of RRs, ORs, WMDs and SMDs using the forest plots.[56] For our meta-analysis, we will run the random-effects model as the statistical model[57 58] that is based on the assumption that the true effect size varies between studies and follows a normal distribution around the mean, which is opposite to the fixed effect model based on the assumption that all studies have the same true effect size.[33] We will calculate the Q statistics and p values for checking the assumption of the homogeneity of effect sizes, and we will determine the $I^2$ statistics for estimating the magnitude of the heterogeneity / variance of the true effect sizes between studies.[33]

We assume that effect size of interventions would vary depending on the follow-period; hence, we will run meta-analyses using combinations of measurements taken at different times such as baseline versus first follow-up measurements, measurements at the first and second follow-up and measurements at the first and the last follow-up. We will run meta-analyses using the Review Manager (RevMan) V.5.3.5 software.[59]

### Assessment of research quality, bias and heterogeneity

Two researchers (SGSS and DN) will independently assess the research quality and any bias in the selected studies using the validated tools (table 3). Any discrepancy or disagreement among the reviewers will be resolved by discussion and consensus between them or through arbitration by the third reviewer (HvW). We will evaluate heterogeneity, that is, variation in study outcomes/effect sizes between studies by the Cochran's Q test with a significance level of $\rho < 0.05$.[33 60] We will calculate $I^2$ statistic[33 61] to determine the proportion of variation in effect size across studies due to heterogeneity considering $I^2$ of 25% as low heterogeneity, 50% moderate heterogeneity[33 61] and 75% as a high heterogeneity/variance between studies.[12 61] In the case of substantial heterogeneity ($I^2 > 50\%$),[49] we will consider running stratified meta-analyses and random-effect meta-regression to ascertain whether effects size was associated with the

methodological or clinical characteristics of the studies included in the meta-analysis.[62] As mentioned above in the Meta-analysis section, we will run the random effects model and report the random effects size and the level of heterogeneity observed in the model. For checking the publication bias in the studies included in our systematic review, we will use two methods: graphical method using funnel plots and statistical method using the Egger's test.[56 63] In assessing the quality of research, we will apply the Grading of Recommendations Assessment, Development and Evaluation approach.[64]

## PATIENT AND PUBLIC INVOLVEMENT

The involvement of patients and public has been suggested in systematic reviews[65]; however, we could not identify any patient diagnosed with loneliness or a suitable member of the public to be involved in the development of this protocol. As such, there was no patient or public involvement at the protocol stage in our study, like other published research protocols.[4 66]

## ETHICS AND DISSEMINATION

This study is a systematic review and meta-analysis of published research. We will neither recruit human participants nor analyse data at an individual participant level; however, we will pool data that will be analysed at the study level. We will therefore not seek ethics approval for this study. We will disseminate our findings through conference papers and presentations and publication of open access articles in peer-reviewed journals.

## CONCLUSION

We believe our research protocol includes a robust search strategy that will enable us to identify primary research studies using digital interventions to reduce loneliness in adults. We believe our robust research methodology and analytical strategy will enable us to meet the objectives of our systematic review and meta-analysis, which will provide the latest evidence on the effectiveness of digital interventions in reducing loneliness among adults. We conclude that appraising the latest empirical research on digital interventions used to reduce loneliness in adults could contribute in informing health and care and public health policy and possibly other stakeholders and private entities (eg, health insurers) aiming to tackle loneliness in the adult population.

**Contributors** All authors were involved in the planning, conception and design of the study. SGSS drafted the manuscript, which was reviewed by DN, HvW and VK. SGSS undertook preliminary literature searches in the PubMed with the help from Bodleian Health Care Libraries, University of Oxford. All authors have read, reviewed, contributed in revising and approved the final manuscript. SGSS is the guarantor. VK supervised the project and obtained research funding for paying the open access article publication charges.

| Table 3 | Research quality and bias assessment tools |
| --- | --- |
| **Study design** | **Research quality and bias assessment tool** |
| Randomised control trials | RoB 2.0 tool (Revised tool for Risk of Bias in Randomized Trials)[67] |
| Non-randomised studies | ROBINS-I tool (Risk Of Bias in Non-randomized Studies—of Interventions)[68] |
| Qualitative Research Studies | JBI Critical Appraisal Checklist for Qualitative Research[69] |

**Funding** This research was funded/supported by the National Institute for Health Research (NIHR) Oxford Biomedical Research Centre. The views expressed are those of the author(s) and not necessarily those of the NHS, the NIHR or the Department of Health.

**Competing interests** None declared.

**Patient consent for publication** Not required.

**Provenance and peer review** Not commissioned; externally peer reviewed.

**ORCID iD**
Syed Ghulam Sarwar Shah http://orcid.org/0000-0002-5713-3686

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
