## [Reviewer comments · BMJ Open]

ARTICLE DETAILS

TITLE (PROVISIONAL)	The effectiveness of digital technology interventions to reduce loneliness in adults: A protocol for a systematic review and meta-analysis
AUTHORS	Shah, Syed Ghulam Sarwar; Nogueras, David; van Woerden, Hugo; Kiparoglou, Vasiliki

VERSION 1 – REVIEW

REVIEWER	Peter J Schulz Institute of Communication & Health, University of Lugano (Università della Svizzera italiana), Switzerland
REVIEW RETURNED	30-Jun-2019

GENERAL COMMENTS	The authors want to conduct a systematic review and meta-analysis to find out which digital technology interventions can be used to treat loneliness in adults. Their plans are state of the art, and carefully thought through. They are also very inclusive, as for instance evidenced by taking in all existing measures of loneliness. The presentation of the plans, especially in the introduction, suffer from some shortcomings, which can easily be edited out: The section Definitions in the INTRODUCTION is highly redundant. The second sentence repeats the first 8lines 24-27), but is not quite as precise as that: the first specifies that desired > real, while the second would also include desired < real, which does not make much sense, but that is not the point. At least half of the fourth sentence (lines 29-33) repeats the first two. And the last two sentences are redundant again as they both state that the health outcomes differ (lines 35-37). Same section defines social isolation “objective feeling” (line 29), which I do not think is a reasonable expression. Line 48, p. 5: an expression like “statistically significantly associated” in this context is unnecessary because are associated only if they are significantly associated. Badly constructed phrases: p. 6, line 6 (“mental health such as depression”, maybe insert “condition”); p. 6, line 11 (“early/premature mortality and all-cause mortality”); Logically, any inclusion criterion can be formulated as an exclusion criterion. Providing each of them in both formulations (p. 8) is unnecessary. You may also want to refer to previous systematic reviews on the topic, among others Chen & Schulz 2016 (JMIR)
--

REVIEWER	Walter Ricciardi 1. Sezione di Igiene, Istituto di Sanità Pubblica, Università Cattolica del Sacro Cuore, Roma, Italia. 2. Department of Woman and Child Health and Public Health - Public Health Area, Fondazione Policlinico Universitario A.Gemelli IRCCS, Roma, Italia
REVIEW RETURNED	12-Jul-2019

GENERAL COMMENTS	This protocol report the planned methodology for the conduction of a systematic review and meta-analysis with the aim to identify and assess the effectiveness of digital technology interventions to reduce the loneliness. The research topic is very current and many efforts should be put in place in order to address it. The pre-defined methodology explained within this protocol is complete and correctly reported. Appropriate instruments available for supporting the development of a systematic review and meta-analysis are cited and properly elaborated for the research. The objectives, methodology, including search strategy, article selection and assessment and data management, and expected results are clearly described. I have just few minor revisions to suggest. Despite this could appears more exhaustive, reporting both the inclusion and the correspondent exclusion criteria seems quite redundant. The authors could avoid to report exclusion criteria that are the exact complement of inclusion criteria. The authors declared that in presence of heterogeneity they will not run the meta-analysis. I would suggest, in addition, to search for possible sources of heterogeneity and then, where possible, to perform stratified meta-analyses. Furthermore, the authors should specify the statistical software/s that they intend to use for the meta-analysis.
--

REVIEWER	Patricia Moreno-Peral Biomedical Research Institute in Malaga (IBIMA)
REVIEW RETURNED	17-Jul-2019

GENERAL COMMENTS	This protocol of systematic review and meta-analysis addresses an interesting topic. The article is well written and the method is systematic and well defined. In the manuscript the authors frame their question very well using the PICO framework. However, I have two main comments: If the evidence is derived from observational studies, a conclusion on causality cannot be provided. In addition, when different types of designs are included, the heterogeneity may be high, which would limit, according to the protocol of the authors, the performance of a meta-analysis. The authors state that they will pool data from studies that will be sufficiently homogeneous. I wonder how the studies are considered sufficiently homogeneous (based on what variables? effect size, population, type of design...). It could be a potential area of bias during the review, so this aspect needs clarification. Minor comments: Do the authors consider that the discussion section is necessary? Why will the authors not contact the original researchers when some of the data are not available in their published articles?
---

	Doing that could avoid excluding studies with missing values from the meta-analysis. Regarding the timeframe of post-intervention, will it only include immediate post-intervention measures? What happens if more than one post-intervention outcome measures have been collected? Please, merge these following sentences: "If the heterogeneity was substantial ($I^2 \geq 50\%$). We would not run the meta-analysis and therefore we will report only to narrative synthesis" (page 12, line 14 and 15). Regarding to this statement, I have two questions: 1) If heterogeneity is < 50, although moderate, will the authors explain that heterogeneity? 2) Have the authors considered, when the heterogeneity is substantial ($I^2 \geq 50\%$), run the meta-analysis and try to explain the heterogeneity through random-effect meta-regression?
--	---

VERSION 1 – AUTHOR RESPONSE

Reviewers actionable advice	Authors reply /action	Location of changes made in the manuscript
Reviewer: 1 [Peter J Schulz]		
the introduction, suffer from some shortcomings		
The section Definitions in the INTRODUCTION is highly redundant. The second sentence repeats the first 8lines 24-27), but is not quite as precise as that: the first specifies that desired > real, while the second would also include desired < real, which does not make much sense, but that is not the point. At least half of the fourth sentence (lines 29-33) repeats the first two. And the last two sentences are redundant again as they both state that the health outcomes differ (lines 35-37).	We thank to the reviewer for pointing out this issue. We have revised the Definitions subsection in the introduction. We have removed the duplication, reduced the text and rewritten the sentences. This sub-section now reads as: In the literature, loneliness and social isolation are often reported together but they differ from each other [1] Loneliness is defined as a subjective feeling that arises when an individual perceives a discrepancy between the actual and the desired social relationships.[2][3] While, social isolation is an objective feeling,[2] which develops due to the absence of social contact with the family, friends,[1] individuals and society[3].In addition, loneliness and social isolation have distinct pathways to adverse health effects.[4,5]Loneliness creates pain and distress[3] and it is associated with adverse health effects.[1]	Page 3, lines 8-14

Same section defines social isolation “objective feeling” (line 29), which I do not think is a reasonable expression.	We have taken this definition from the literature. There are many studies and reports that define social isolation as an objective feeling for example see references 1, 2, 3.	Page 3, lines 11-12
Line 48, p. 5: an expression like “statistically significantly associated” in this context is unnecessary because are associated only if they are significantly associated.	We have revised this sentence by deleting ‘statistically significantly’ from the sentence. It now reads as: In older adults, loneliness is associated with female gender,.....	Page 3, lines 23-24
Badly constructed phrases: p. 6, line 6 (“mental health such as depression”, maybe insert “condition”);	We have added the word ‘conditions’ in this sentence. It now reads as: Loneliness is associated with mental health conditions such as depression, low self-esteem, anxiety, perceived stress,.....	Page 4, lines 3-4
p. 6, line 11 (“early/premature mortality and all-cause mortality”);	Early mortality and premature mortality are synonymously used in the literature. Therefore we wrote them together. However, we take the term ‘early’ out and report only ‘premature’ in this sentence. It reads as follows: ... but also increases risk of premature mortality and all-cause mortality.....	Page 4, lines 6-8
Logically, any inclusion criterion can be formulated as an exclusion criterion. Providing each of them in both formulations (p. 8) is unnecessary.	Many thanks for raising this issue. We have removed the exclusion criteria, from the text and Table 1.	Page 7, lines 27-28 and Table 1 on pages 7-8
You may also want to refer to previous systematic reviews on the topic, among others Chen & Schulz 2016 (JMIR).	Again, many thanks for suggesting a systematic review by Chen & Schulz 2016, which we have read and found useful but not very relevant at the protocol stage of our study.	No action
Reviewer: 2 (Walter Ricciardi)		
Despite this could appears more exhaustive, reporting both the inclusion and the correspondent exclusion criteria seems quite redundant. The authors could avoid to report exclusion criteria that are the exact complement of inclusion criteria.	Many thanks for raising this issue. We have removed the exclusion criteria, from the text and Table 1.	Page 7, lines 27-28 and Table 1 on pages 7-8
The authors declared that in presence of heterogeneity they will not run the meta-analysis. I would suggest, in addition, to search for possible sources of heterogeneity and then, where possible, to perform stratified meta-analyses.	We agree with the reviewer’s suggestion and we have included consideration of undertaking stratified meta-analyses in the case of substantial heterogeneity ($I^2 > 50\%$).	Page 13, lines 29-32

	We have incorporated this suggestion in the revised text as follows: In the case of substantial heterogeneity ($I^2 > 50\%$), [49] we will consider running stratified meta-analyses and random-effect meta-regression to ascertain whether effects size was associated with the methodological or clinical characteristics of the studies included in the meta-analysis. [62]	
Furthermore, the authors should specify the statistical software/s that they intend to use for the meta-analysis.	This is a critical issue raised by this reviewer. We will use the RevMan 5 software for performing the meta-analysis. We add the following statement: We will run meta-analyses using the Review Manager (RevMan), version 5.3.5, software [59]	Page 2 -abstract, lines 26-27) and page 13, lines 19-20
Reviewer: 3 (Patricia Moreno-Peral)	.	
If the evidence is derived from observational studies, a conclusion on causality cannot be provided.	Observational studies include case control and cohort studies (before and after studies). We will include these types of studies that use a digital technology intervention. However, we will not include cross sectional and evaluation studies. Which we have removed from the inclusion criteria.	Page 6-7 (Table 1, parameter: study types/designs)
In addition, when different types of designs are included, the heterogeneity may be high, which would limit, according to the protocol of the authors, the performance of a meta-analysis.	We will consider undertaking stratified meta-analyses in the case of high heterogeneity ($I^2 > 50\%$).	Page 13, lines 29-32
The authors state that they will pool data from studies that will be sufficiently homogeneous. I wonder how the studies are considered sufficiently homogeneous (based on what variables? effect size, population, type of design...). It could be a potential area of bias during the review, so this aspect needs clarification.	We have revised this statement to avoid any confusion and facilitating meta-analysis. We have revised this sentence as follows: We will extract and analyse data at the study level.	Page 12, lines 22-23
Minor comments:		
Do the authors consider that the discussion section is necessary?	Discussion section is not essential but we included it because some published protocols had this section included. However, we have removed the discussion section in our revised manuscript as suggested by the reviewers.	Page 14 (lines-13-23)

Why will the authors not contact the original researchers when some of the data are not available in their published articles? Doing that could avoid excluding studies with missing values from the meta-analysis.	We had included this statement because we are not sure whether writing to authors of published articles for data would be successful, especially because a lot of academics and researchers change their jobs and employer institutions. However, we have changed our mind and we will attempt to write to the authors on included studies for additional information regarding data published in their articles. We have revised our statement as follows: If data reported in selected studies were found difficult to extract or incomplete, we will not attempt to contact the original researchers with regard to any queries regarding the data reported in their published articles.	Page 12, lines 6-8
Regarding the timeframe of post-intervention, will it only include immediate post-intervention measures? What happens if more than one post-intervention outcome measures have been collected?	Thanks for raising this point, which we have clarified this issue by adding the following text at various points in the manuscript as follows: Timing and effect measures: We will extract information on measurement of loneliness at the baseline (before the introduction of the intervention) and every follow-up measurement for both the intervention groups and control group depending on the design of studies included in our systematic review. In the case of more than one follow-up measurement, we will run a series of meta-analyses as explained in the meta-analysis section below. Data extraction process section: We will extract data reported for all measurements including the baseline measurement and all subsequent follow-up measurements. In addition, we have included an additional column in the data extraction template (Table 3) to capture data measurement at various points in the studies	Page 7, lines 20-24 Page 11, lines 36-37 Page 12 , Table 3

Please, merge these following sentences: “If the heterogeneity was substantial ($I^2 \geq 50\%$). We would not run the meta-analysis and therefore we will report only to narrative synthesis” (page 12, line 14 and 15).	Thanks. We have merged this sentence by removing a full stop before the word ‘we would’.	Page 13 , lines 29-30
Regarding to this statement, I have two questions: 1) If heterogeneity is < 50, although moderate, will the authors explain that heterogeneity? 2) Have the authors considered, when the heterogeneity is substantial ($I^2 \geq 50\%$), run the meta-analysis and try to explain the heterogeneity through random-effect meta-regression?	1. Yes, we will report the observed heterogeneity levels in the meta-analyses. 2. Thanks for this suggestion. As stated above, we will run stratified meta-analyses and meta-regression to check the influence of study characteristics on the effect sizes. In this regard we have added the following statement: In the case of substantial heterogeneity ($I^2 > 50\%$),[49] we will consider running stratified meta-analyses and random-effect meta-regression to ascertain whether effects size was associated with the methodological or clinical characteristics of the studies included in the meta-analysis.[62]	Page 13, lines 29-32

VERSION 2 – REVIEW

REVIEWER	Peter J Schulz Institute of Communication & Health, University of Lugano (università della Svizzera italiana), Switzerland
REVIEW RETURNED	20-Aug-2019
GENERAL COMMENTS	My review of the first version of this manuscript pointed out a handful of stylistic elements I found had to be improved. That the authors did in a satisfactory manner. Other changes, which were more comprehensive than my suggestions, do not contain anything that would alter my favorable assessment of the earlier version. I support publication.
REVIEWER	Walter Ricciardi 1. Sezione di Igiene, Istituto di Sanità Pubblica, Università Cattolica del Sacro Cuore, Roma, Italia; 2. Department of Woman and Child Health and Public Health - Public Health Area, Fondazione Policlinico Universitario A. Gemelli IRCCS, Roma, Italia;
REVIEW RETURNED	11-Aug-2019
GENERAL COMMENTS	The Authors addressed properly the suggested revisions and provided clear responses to my comments. I have no further comments. My recommendation is to accept this revised version.
REVIEWER	Patricia Moreno-Peral IBIMA. Spain.
REVIEW RETURNED	29-Aug-2019
GENERAL COMMENTS	The authors have addressed my comments. However, I still question the use of case control and cohort studies with the aim to know the effectiveness of digital technology interventions to reduce loneliness